# LEARNING A DIFFUSION MODEL POLICY FROM REWARDS VIA Q-SCORE MATCHING

## ABSTRACT

Diffusion models have become a popular choice for representing actor policies in behavior cloning and offline reinforcement learning. This is due to their natural ability to optimize an expressive class of distributions over a continuous space. However, previous works fail to exploit the score-based structure of diffusion models, and instead utilize a simple behavior cloning term to train the actor, limiting their ability in the actor-critic setting. In this paper, we focus on off-policy reinforcement learning and propose a new method for learning a diffusion model policy that exploits the linked structure between the score of the policy and the action gradient of the Q-function. We denote this method *Q-score matching* and provide theoretical justification for this approach. We conduct experiments in simulated environments to demonstrate the effectiveness of our proposed method and compare to popular baselines. Our code is publicly available at https://www.scorematchingrl.com.

## 1 INTRODUCTION

Reinforcement Learning (RL) has firmly established its importance across a range of complex domains, from discrete game environments like Go, Chess, and Poker (Silver et al., 2016; 2017; Brown & Sandholm, 2019) to continuous environments like goal-oriented robotics (Kober et al., 2013; Sünderhauf et al., 2018; Ibarz et al., 2021; Wu et al., 2022). Robotics RL applications typically need to work in a continuous vector space for both states and actions. This not only makes traditional RL algorithms designed for discrete state and action spaces infeasible, but makes parameterizing the policy (distribution of actions) a difficult challenge, where one must typically choose between ease of sampling (e.g. Gaussians (Agostini & Celaya, 2010)) and expressiveness.

Recently, diffusion models (Hyvärinen & Dayan, 2005; Ho et al., 2020) have emerged as a promising avenue for parameterizing distributions. These models, rooted in the idea of iterative increments of noising and denoising from a distribution, have shown great potential in generative tasks (Rombach et al., 2022; Watson et al., 2022). In the context of RL, diffusion models offer both expressiveness and easy sampling, since normalization constants do not need to be computed for sampling. However, their adoption in RL is relatively nascent, and the nuances of their implementation and performance are still subjects of investigation.

One unexplored approach is through the alignment of the learned policy's score, denoted by $\nabla_a \log(\pi(a|s))$, with the score of an optimal policy, denoted by $\nabla_a \log(\pi^*(a|s))$. However, traditional score matching is ill-posed in this setting, because we not only lack samples from $\nabla_a \log(\pi^*(a|s))$, but also from $\pi^*(a|s)$ itself. Our primary result emphasizes that in the context of RL, one can match the score of $\pi$ to that of $\pi^*$ by iteratively matching the score of $\pi$ to the gradient of the state-action value function with respect to action, $\nabla_a Q^\pi(s,a)$. This offers a new, geometric perspective on policy optimization, where the focus for policy optimization becomes iteratively pushing the vector field $\nabla_a \log(\pi(a|s))$ towards the vector field $\nabla_a Q^\pi(s,a)$. We call this approach *Q-score matching* (QSM).

We then use this novel method on off-policy reinforcement learning scenarios, an important but yet unexplored area for diffusion model policies. Without a fixed distribution to sample from (akin to what is given for behavior cloning or offline RL applications), it is unclear how exactly to train a

diffusion model represented policy. We postulate and empirically demonstrate that QSM is a viable algorithm for learning diffusion model policies.

The paper is structured as follows: we begin by establishing a continuous-time formulation of RL via stochastic differential equations to allow score-based analysis while keeping the stochastic flexibility of the RL setting. We then introduce the standard policy gradient for diffusion models in this setting, and afterwards give a theoretical introduction to QSM. Finally, we lay out our empirical framework, results, and the broader implications of our method for learning diffusion model policies in reinforcement learning problems.

## 1.1 RELATED WORK

We now introduce the relevant related work for this paper. The most direct relation is the line of work related to diffusion models in the RL setting, which is relatively new but actively explored. We discuss some particular works related to various parts of this paper's setting, including the action gradient of the Q-function and the score of the policy distribution.

### 1.1.1 DIFFUSION MODELS IN REINFORCEMENT LEARNING

Diffusion models, characterized by their incremental noise-driven evolution of data, have found various applications in RL, ranging from imitating expert behavior to optimizing policies using complex action-value functions. The following are some notable examples:

**Behavior cloning:** "Behavior cloning" is a type of imitation learning where an agent tries to mimic the behavior of an expert without explicit reward feedback. Much of the earlier work in diffusion models for policy learning has been for behavior cloning (Janner et al., 2022; Reuss et al., 2023), as the specifics of the behavior cloning setting (matching a distribution to a given dataset) fit more closely to the original design principle of diffusion models, namely through score matching (Hyvärinen & Dayan, 2005). In a similar vein, work has been done on learning a stochastic state dynamics model using a diffusion model (Li et al., 2022a).

**Offline Q-learning:** Offline RL techniques leverage existing datasets to learn optimal policies without further interaction with the environment. Similar to the behavior cloning setting, optimizing the policy using a large fixed dataset fits more closely with the common usage of diffusion models for learning distributions, and in combination with the above policy gradient formula for diffusion models has enabled many recent works in this area (Wang et al., 2022; Suh et al., 2023; Kang et al., 2023; Hansen-Estruch et al., 2023; Lu et al., 2023).

**Policy gradient:** Policy gradient methods seek to directly optimize the policy by computing gradients of the expected reward with respect to the policy parameters (Sutton et al., 1999). Previous work has derived a formula for the policy gradient with respect to a diffusion model's parameters (Black et al., 2023), but such formulas are much more general and do not fully exploit the structure of a diffusion model. For example, the new expectation for the policy gradient becomes dependent on internal action samples, making the estimates less sample efficient (see Section 3).

There are additional works that use the action gradient of the Q-function for learning (Silver et al., 2014; Berseth et al., 2018; D'Oro & Jaśkowski, 2020; Li et al., 2022b; Sarafian et al., 2021), where the standard policy gradient is expanded to include the action gradient of the Q-function through the chain rule, but such methods require an explicit representation for the full policy distribution $\pi(a|s)$, which is not readily available for diffusion models.

**Diffusion-QL:** Although Wang et al. (Wang et al., 2022) perform experiments in the offline-RL setting with a behavior cloning term, they propose a method for pure Q-learning: training on $Q$ itself as the objective, and backpropagating through the diffusion model evaluation. However, such training still does not utilize the unique structure of diffusion models and presents computational challenges (e.g. exploding/vanishing gradients from differentiating through model applied on itself).

### 1.1.2 STOCHASTIC OPTIMAL CONTROL

At a conceptual level, our work is rooted in the principles of stochastic optimal control (Fleming & Rishel, 2012; Kirk, 2004; Bellman, 1954), which deals with optimizing systems subjected to random disturbances over time. Especially relevant to our context is the continuous-time formulation,

where the control strategies are adjusted dynamically in response to evolving system states. However, much of stochastic optimal control literature typically assumes access to some model of the state dynamics. Instead of assuming a state dynamics model, we assume a model for the expected discounted rewards over time (the Q-function), leading to new motivations for the theoretical development of our method. Nonetheless, exploring the link between this paper and stochastic optimal control is an interesting direction for future work, in particular given the surface similarities between Theorems 1 and 2 and the Hamilton-Jacobi-Bellman equation (Bellman, 1954).

## 2 PROBLEM FORMULATION

We now define and introduce core mathematical notation and objects used. Of notable difference from a standard reinforcement learning formulation is the use of a continuous time formulation, which is helpful to simplify the theoretical statements of this paper.

### 2.1 NOTATION

We first introduce non-standard notation used throughout the main body and proofs in the appendix.

1. Different notations for the time derivative of a path $x(t)$ are used, depending on the setting. In the non-stochastic/deterministic setting, we use dot notation $\dot{x}(t) := \frac{d}{dt}x(t)$. In the stochastic setting, we use the standard SDE notation, using $dx(t)$ instead of $\frac{d}{dt}x(t)$.

2. In the main body and proofs, we often refer to the "score" of the action distribution, which is always denoted $\Psi$ and always used as the vector field defining the flow of actions over time. Not all settings have $\Psi$ line up with the classical score of a distribution, but we use the terminology "score" throughout to highlight the analogy to a distribution's true score $\nabla_a \log \pi(a|s)$, as seen most clearly through the Langevin dynamics with respect to a distribution's score (Langevin, 1908; Papanicolaou, 1977; Welling & Teh, 2011).

### 2.2 DEFINITIONS

We denote the *state space* as a Euclidean space $\mathcal{S} = \mathbb{R}^s$, and the *action space* as another Euclidean space $\mathcal{A} = \mathbb{R}^a$. For the theoretical portion of this paper, we consider the following stochastic, continuous-time setting for state and action dynamics:

$$
\begin{aligned}
ds &= F(s,a)dt + \Sigma_s(s,a)dB_t^s, \\
da &= \Psi(s,a)dt + \Sigma_a(s,a)dB_t^a, \\
s(0) &= s_0, \\
a(0) &= a_0.
\end{aligned}
\tag{1}
$$

$\Psi : \mathcal{S} \times \mathcal{A} \to \mathcal{A}$ corresponds to the "score" of our policy and is the main parameter for policy optimization in this setting. $F : \mathcal{S} \times \mathcal{A} \to \mathcal{S}$ corresponds to the continuous state dynamics, and $\Sigma_s(s,a), \Sigma_a(s,a)$ are functions from $\mathcal{S} \times \mathcal{A}$ to positive semidefinite matrices in $\mathbb{R}^{s \times s}$ and $\mathbb{R}^{a \times a}$ respectively, corresponding to the covariance structure for the uncertainty in the dynamics of $s(t)$ and $a(t)$. The covariances are with respect to the separate Brownian motions $B^s, B^a$, each embedded in $\mathcal{S}$ and $\mathcal{A}$ respectively.

Our main objective in this paper is to maximize path integral loss functions of the following form:

$$
Q^\Psi(s_0, a_0) = \mathbb{E} \int_0^\infty \gamma^t r(s(t, s_0, a_0))dt,
\tag{2}
$$

where $r : \mathcal{S} \to [0,1]$ is the *reward function*, the expectation is taken over the stochastic dynamics given in equation 1, $s(t, s_0, a_0)$ is a sample of the path $s(\cdot)$ at time $t$ from initial conditions $(s_0, a_0)$, and $\gamma \in (0,1)$ is a fixed constant corresponding to a "discount factor". Discretizing by time gives a more familiar formula for the Q-function:

$$
Q^\Psi(s_0, a_0) = \mathbb{E} \sum_{t=0}^\infty \gamma^t r(s(t, s_0, a_0)),
\tag{3}
$$

and furthermore if we cut off at some horizon where $\sum_{T+1}^{\infty} \gamma^i \approx 0$:

$$Q^{\Psi}(s_0, a_0) = \mathbb{E} \sum_{t=0}^{T} \gamma^t r(s(t, s_0, a_0)). \tag{4}$$

We write superscript $\Psi$ because we want to consider $Q$ as a function not of the initial conditions, but of the score $\Psi$, and try to find a score $\Psi^*$ that maximizes $Q^{\Psi}(s_0, a_0)$ for a fixed initial condition (or an expectation over a distribution of initial conditions).

There is of course motivation for optimizing the noise covariance structure for the actions $\Sigma_a$, but we save this optimization for a future work and here purely focus on the score $\Psi$.

**Our objective** in this paper is then to maximize the following function with respect to the vector field $\Psi : \mathcal{S} \times \mathcal{A} \to \mathcal{A}$ for some distribution over initial state/action pairs $\mathbb{P} \times \pi$:

$$J(\Psi) = \mathbb{E}_{\mathbb{P} \times \pi} Q^{\Psi}(s, a). \tag{5}$$

## 2.3 TIME DISCRETIZATION

One may observe that the action model in equation 1 is not standard, since actions $a(t)$ are modeled as a smooth flow over time, rather than an explicit function of the state $s(t)$. The motivation for this model comes when we discretize in time via the Euler-Maruyama method, and further when we discretize at different time scales with respect to the state and action dynamics in the following way:

$$s_{t+1} = s_t + F(s_t, a_t) + z, z \sim \mathcal{N}(\mathbf{0}, \Sigma_s(s_t, a_t)),$$
$$a_t^i = a_t^{i-1} + \frac{1}{K}\Psi(s_t, a_t^{i-1}) + z, z \sim \mathcal{N}(\mathbf{0}, \frac{1}{K}\Sigma_a(s_t, a_t^{i-1})), \tag{6}$$
$$a_{t+1} = a_t^K,$$

where $a_t^i := a_{t+i/K}$. By discretizing the action dynamics at $K$ times the fidelity of the state dynamics discretization, we recover a time-invariant diffusion model of depth $K$, with denoising mean represented by $\Psi$ and variance represented by $\Sigma_a$. As a result, we expect the theory on the model in equation 1 to also approximately hold for systems of the form given in equation 6 where actions are represented by a diffusion model, up to the error incurred by time discretization.

## 3 POLICY GRADIENT FOR DIFFUSION POLICIES

Recall that the policy gradient, the gradient of the global objective $J(\theta) = \mathbb{E}_{s,a} Q^{\pi}(s, a)$ is given by the following (Sutton et al., 1999):

$$\nabla_\theta J(\theta) = \mathbb{E}_{(s_0, a_0, s_1, a_1, \dots)} \sum_{t=1}^{\infty} Q(s_t, a_t) \nabla_\theta \log \pi_\theta(a_t | s_t), \tag{7}$$

where the distribution of the expectation is with respect to the policy $\pi$ and the environment dynamics. For a time-discretization of a diffusion model policy, we do not have access to global probability $\pi(a|s)$, but rather just the incremental steps $\pi(a^\tau | a^{\tau-1}, s)$ (superscript here indicates internal diffusion model steps), and would need to integrate over all possible paths to get the following:

$$\pi(a^K | s) = \int_{a^{K-1}} \cdots \int_{a^1} \pi(a^1 | s) \prod_{\tau=2}^{K} \pi(a^\tau | a^{\tau-1}, s) da^1 \cdots da^{K-1}, \tag{8}$$

which is quickly infeasible to compute with respect to $K$. However, after substituting equation 8 into equation 23, we can simplify and obtain the following form:

$$\nabla_\theta J(\theta) = \mathbb{E}_{(s_0, \{a_0^\tau\}_{\tau=1}^K, s_1, \dots)} \sum_{t=1}^{\infty} Q(s_t, a_t^K) \left( \sum_{\tau=1}^{K} \nabla_\theta \log \pi_\theta(a_t^\tau | a_t^{\tau-1}, s_t) \right). \tag{9}$$

Proof of this can be found in Appendix B.1.

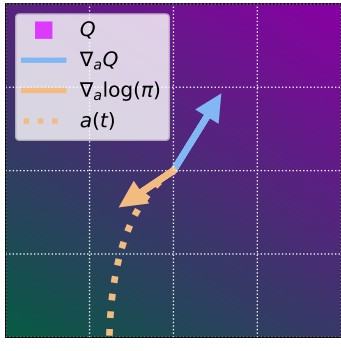

(a) Random initialized $\Psi^0$.

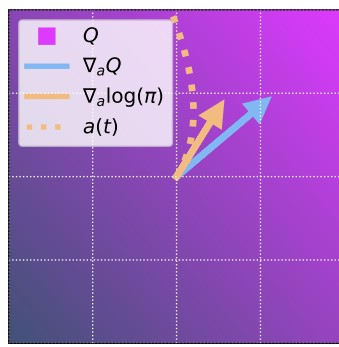

(b) One step of QSM $\Psi^1$.

Figure 1: A visual description of Theorem 1 and Theorem 2, and the implied update rule for a policy $\pi$ parameterized by a diffusion model. If there is any discrepancy between the score $\nabla_a \log(\pi(a|s))$ (orange vector, denoted $\Psi$ in the paper and optimized directly) and the action gradient $\nabla_a Q(s, a)$ (blue vector), we can forcefully align the score to the $Q$ action gradient to strictly increase the $Q$ value at $(s, a)$.

## 4   POLICY OPTIMIZATION VIA MATCHING THE SCORE TO THE Q-FUNCTION

We now introduce the main theoretical contribution of this work: an alternative method for updating the policy given an estimate of the $Q$-function that avoids the aforementioned problems from policy gradients.

The theory of this section is relevant for continuous-time dynamical systems. Thus, we consider the following continuous-time $Q$-function:

$$Q^\Psi(s_0, a_0) = \mathbb{E}\left[\int_0^\infty \gamma^t r(s(t, s_0, a_0))\right], \tag{10}$$

and the corresponding total energy function $J(\Psi)$:

$$J(\Psi) = \mathbb{E}_{(s,a)} Q^\Psi(s, a), \tag{11}$$

where the expectation for $Q^\Psi$ is taken over sampled paths of $(s(t), a(t))$ with respect to the stochastic dynamics from equation 1 starting from $(s_0, a_0)$. In computation, we can also consider a finite-dimensional parameterization $\Psi_\theta$ for parameters $\theta \in \mathbb{R}^d$ and the parameterized loss:

$$J(\theta) = \mathbb{E}_{(s,a)} Q^{\Psi_\theta}(s, a), \tag{12}$$

where the parameters $\theta$ parameterize the vector field $\Psi$ for the actions given in equation 1.

Our goal is to match the score $\Psi$ with that of an optimal distribution with respect to $J$, denoted $\Psi^*$. However, unlike the standard score matching literature, we not only do not have access to any samples of $\Psi^*$, but not even from the action distribution $\pi^*$ it generates. Thus, our matching approach will require access to a surrogate $\Psi^*$ approximator.

One hypothesis from dimensional analysis is to compare $\Psi$ to $\nabla_a Q^\Psi$, which is also a vector field from $\mathcal{S} \times \mathcal{A}$ to $\mathcal{A}$. Intuitively, $\nabla_a Q^\Psi$ provides which ways actions should be modified locally to maximize expected reward. As such, we define $\Psi^* \approx \nabla_a Q$ and define our actor update as an iterative optimization of the score $\Psi$ against the $\nabla_a Q$ target.

One can optimize $\Psi$ by iteratively matching it to the action gradient of its $Q$-function, $\nabla_a Q^\Psi(s, a)$. Our theoretical statements for this are captured in Theorems 1 and 2, and their proofs in appendix B.2 and B.3 respectively.

### 4.1   NON-STOCHASTIC SETTING

We begin by proving the above result in the non-stochastic setting. That is, the state and action both evolve as paired ordinary differential equations. To further simplify, we will begin by trying to

optimize just the $Q$-function at a fixed point $Q(s_0, a_0)$, or without loss of generality of the theory, the $Q$-function at the origin $Q(\mathbf{0}, \mathbf{0})$. This is still quite a rich setting, since even for a fixed point $Q(s, a)$ is still a path integral equation. The following theorem formalizes the setting, and gives a concrete statement relating the optimal policy's score to the action gradient of the $Q$-function.

**Theorem 1** (Optimality condition, deterministic setting). *Consider the following joint deterministic dynamics governing the state $s(t) \in \mathbb{R}^s$ and action $a(t) \in \mathbb{R}^a$:*

$$\dot{s}(t) = F(s(t)), \quad \dot{a}(t) = \Psi(s(t), a(t)),$$
$$s(0) = \mathbf{0}, \quad a(0) = \mathbf{0}. \tag{13}$$

*where $s(t) \in \mathbb{R}^s$, $a(t) \in \mathbb{R}^a$, $\|\Psi(s, a)\|_2 \leq C$ for all $(s, a)$, and $\Psi$ is Lipschitz with respect to $\|\cdot\|_2$. Denote $s(t, s_0, a_0)$ the resulting state $s(t)$ from initial conditions $s_0, a_0$. Let $r : \mathbb{R}^s \to [0, 1]$ be a smooth function. Finally, let $\Psi^* : \mathbb{R}^s \times \mathbb{R}^a \to \mathbb{R}^a$ be a vector field that maximizes the following path norm for fixed $\gamma \in (0, 1)$ over smooth flows $\Psi$ with norm bound $C$ and bounded Lipschitz constant:*

$$J(\Psi) = Q^\Psi(\mathbf{0}, \mathbf{0}), \tag{14}$$

$$Q^\Psi(s_0, a_0) = \int_0^\infty \gamma^t r(s(t, s_0, a_0)) dt, \tag{15}$$

*it follows that $\Psi^*(s, a) = \alpha_{s,a} \nabla_a Q^{\Psi^*}(s, a)$ for some $\alpha_{s,a} > 0$ for all $(s, a)$ along the trajectory $(s(t), a(t))$ where $\nabla_a Q^{\Psi^*}(s, a) \neq \mathbf{0}$.*

Proof can be found in Section B.2. *At their core, Theorems 1 and 2 state the following*: if there is anywhere that $\Psi(s, a)$ and $\nabla_a Q^\Psi(s, a)$ are not aligned, then a new vector field $\Psi'$ resulting from aligning $\Psi(s, a)$ with $\nabla_a Q^\Psi(s, a)$ will strictly increase $Q^{\Psi'}(\mathbf{0}, \mathbf{0})$ towards $Q^*(\mathbf{0}, \mathbf{0})$. As one can imagine, once we extend the above to losses of integrals over $Q$, the collinear condition extends to the entire integrated space.

**Corollary 1.** *Consider the setting of Theorem 1, but with the following modified loss:*

$$J(\Psi) = \mathbb{E}_{\mathbb{P} \times \pi} Q^\Psi(s, a), \tag{16}$$

*where $\mathbb{P} \times \pi$ is a probability measure over $\mathbb{R}^s \times \mathbb{R}^a$ absolutely continuous with respect to the Lebesgue measure. For any maximizer $\Psi^*$, it follows that $\Psi^*(s, a) = \alpha_{s,a} \nabla_a Q^{\Psi^*}(s, a)$ for all $(s, a) \in \mathbb{R}^s \times \mathbb{R}^a$ where $\nabla_a Q^{\Psi^*}(s, a) \neq \mathbf{0}$.*

## 4.2 STOCHASTIC SETTING

We now extend to stochastic differential equations over the actions, as modeled by equation 1. Fortunately, the theory for the deterministic case in Theorem 1 extends rather identically to the fully stochastic case.

**Theorem 2** (Optimality condition, stochastic setting). *Consider the following joint stochastic dynamics governing the state $s(t) \in \mathbb{R}^s$ and action $a(t) \in \mathbb{R}^a$:*

$$ds(t) = F(s(t), a(t))dt + \Sigma_s(s(t), a(t))dB_t^s,$$
$$da(t) = \Psi(s(t), a(t))dt + \Sigma_a(s(t), a(t))dB_t^a,$$
$$s(0) = s_0, \tag{17}$$
$$a(0) = a_0,$$

*where $\Psi, F$ are globally Lipschitz functions defined from $\mathbb{R}^s \times \mathbb{R}^a$ to $\mathbb{R}^a$, $\Sigma_s(s, a)$ is a globally Lipschitz function from $\mathbb{R}^s \times \mathbb{R}^a$ to positive semidefinite matrices in $\mathbb{R}^{s \times s}$ (and similarly for $\Sigma_a(s, a)$), and $B_t^s, B_t^a$ are separate standard Brownian motions in $\mathbb{R}^s$ and $\mathbb{R}^a$. Further consider the following loss function with respect to the vector field $\Psi$:*

$$J(\Psi) = \mathbb{E}_{\mathbb{P} \times \pi} Q^\Psi(s, a), \tag{18}$$

$$Q^\Psi(s_0, a_0) = \mathbb{E} \int_0^\infty \gamma^t r(s(t, s_0, a_0)) dt, \tag{19}$$

*where $\mathbb{P} \times \pi$ is a probability measure over $\mathbb{R}^s \times \mathbb{R}^a$, and the expectation for $Q^\Psi$ is taken over sampled paths of $(s(t), a(t))$ starting from $(s_0, a_0)$. For any optimal vector field $\Psi^*$ with respect to $J$, it follows that $\Psi^*(s, a) = \alpha_{s,a} \nabla_a Q^{\Psi^*}(s, a)$ for some $\alpha_{s,a} > 0$ for all $(s, a) \in \mathbb{R}^s \times \mathbb{R}^a$ where $\nabla_a Q^{\Psi^*}(s, a) \neq \mathbf{0}$ in the support of the distribution generated by $\mathbb{P}, \pi$, and the stochastic dynamics of $(s(t), a(t))$.*

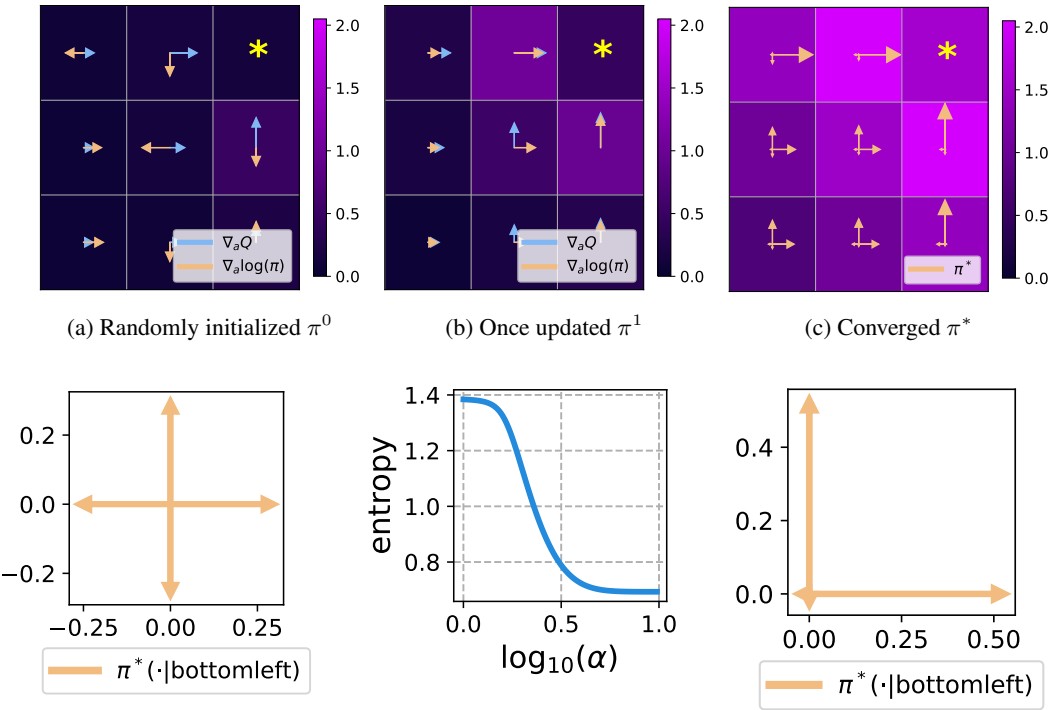

(a) Randomly initialized $\pi^0$      (b) Once updated $\pi^1$      (c) Converged $\pi^*$

Figure 2: Pedagogical simulation of our algorithm's reduction to a simple single-goal gridworld setting. **The top row** is a visualization of two iterates of $\pi(a|s) \leftarrow e^{\alpha Q^\pi(s,a)} / \sum_{a'} e^{TQ^\pi(s,a')}$, for $\alpha = 2$. The color of each square is the expected reward starting from that square, and we use the local maximizing direction to define discrete gradients: $\nabla_a Q(s,a) := a^* Q(s, a^*)$, where $a^* := \arg\max_{a'} Q(s, a')$, and similarly for $\nabla_a \log(\pi(a|s))$. **The bottom row** shows the effect of the parameter $\alpha$ on the entropy of the converged distribution $\pi * (s|a)$.

Proof can be found in Appendix B.3. While this theorem does not imply a specific algorithm, it does provide theoretical justification for a class of policy update methods: iteratively matching $\Psi(s,a)$ to $\nabla_a Q(s,a)$ will provide strict increases to the resulting Q-function globally.

### 4.3 PEDAGOGICAL REDUCTION IN GRIDWORLD

To provide intuition for Theorem 2, we illustrate a reduction of the theorem in gridworld, where the state space $\mathcal{S} = \{1, \ldots, M\} \times \{1, \ldots, N\}$ for some fixed $M, N \in \mathbb{N}$, and $\mathcal{A} = \{\text{LEFT}, \text{RIGHT}, \text{UP}, \text{DOWN}\}$. While diffusion models are not well-defined in discrete space, there is a sensible reduction to gridworld through the Langevin dynamics link.

Suppose (for Euclidean $\mathcal{S}$ and $\mathcal{A}$) we have completed the matching described in Theorem 2 and found a score $\Psi^*$ such that $\Psi^*(s,a) = \alpha \nabla_a Q(s,a)$ for some fixed $\alpha > 0$. Consider the dynamics of equation 1 for a fixed state ($F(s,a) = 0, \Sigma_s(s,a) = 0$), and setting $\Sigma_a(s,a) = \sqrt{2}I_a$; given certain conditions (Bakry et al., 2014), we have the following result for the stationary distribution of actions $a(t)$ as $t \to \infty$ for any fixed $s \in \mathcal{S}$, denoted $\pi(a|s)$:

$$\pi(a|s) \sim e^{\alpha Q(s,a)}. \tag{20}$$

Thus, we can view QSM as matching the full action distribution $\pi(a|s)$ the Boltzmann distribution of the Q-function, $\frac{1}{Z} e^{\alpha Q(s,a)}$, where $Z = \int_\mathcal{A} e^{\alpha Q(s,a)}$. While directly setting $\pi(a|s) \sim e^{\alpha Q(s,a)}$ is infeasible in continuous state/action spaces, we can represent the probabilities $\pi(a|s)$ directly as a matrix of shape $|\mathcal{S}| \times |\mathcal{A}|$ in the finite case and use the following update rule:

$$\pi'(s|a) = \frac{e^{\alpha Q^\pi(s,a)}}{\sum_\mathcal{A} e^{\alpha Q^\pi(s,a)}}. \tag{21}$$

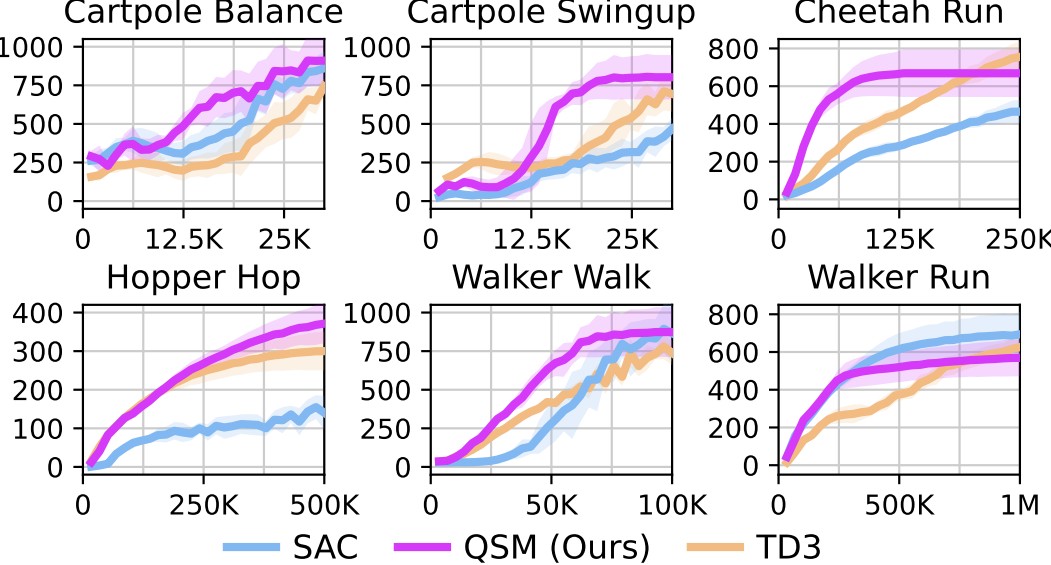

Figure 3: Experimental results across a suite of six continuous control tasks. QSM matches and sometimes outperforms TD3 and SAC performance on the tasks evaluated, particularly in samples needed to reach high rewards. We find that policies trained via QSM learn rich action distributions that provide informative action samples, whereas TD3 and SAC are constrained to draw samples from normal and Tanh normal distributions, respectively.

This update rule is the same as *soft policy iteration* (Haarnoja et al., 2018), but using the standard Q-function rather than the soft Q-function. Nonetheless, we still see in simulated gridworld environments that $\alpha$ acts as an inverse entropy regularization parameter: the lower $\alpha$ is, the higher the entropy of the converged distribution $\pi^*(a|s)$. We visualize this update rule in a simple gridworld environment, depicted in fig. 2.

## 5 EXPERIMENTS

In this section, we describe a practical implementation of QSM, and evaluate our algorithm on various environments from the Deepmind control suite (Tunyasuvunakool et al., 2020). We seek to answer the following questions:

1. Can QSM learn meaningful policies provided limited interaction with the environment?
2. Does QSM learn complex, multi-modal policies?
3. How does QSM compare to popular baselines?

In particular, the first point allows us to verify that using the score-based structure of diffusion models allows us to train diffusion model policies in a sample-efficient manner. We implement QSM using N-step returns (Sutton & Barto, 2018) and the Q-function optimization routine outlined in DDPG (Lillicrap et al., 2019). For each update step, actions from the replay buffer are noised according to a variance-preserving Markov chain, similar to Hansen-Estruch et al. (2023). We then evaluate $\nabla_a Q$ and provide this as the target to our score model $\Psi$. This update is computed for every environment step with batches of size 256 sampled from the replay buffer. Both the critic and score model are parameterized as 2-layer MLPs with 256 units per layer. Pseudocode for the algorithm is provided in algorithm 1. In this work, we add a small amount of Gaussian noise to the final sampled actions. We note that other, interesting exploration strategies exist, such as limiting the amount of denoising steps applied to the action. However, in this work we focus mainly on QSM, and leave additional study of exploration to future work.

### 5.1 CONTINUOUS CONTROL EVALUATION

---

**Algorithm 1** Q-Score Matching (QSM)

---

Initialize critic networks $Q_{\theta_1}$, $Q_{\theta_2}$, and score network $\Psi_\phi$ with random parameters $\theta_1$, $\theta_2$, $\phi$
Initialize target networks $\theta_1' \leftarrow \theta_1$, $\theta_2' \leftarrow \theta_2$
Initialize replay buffer $\mathcal{B}$
**while** *not converged* **do**

    Choose action by iteratively denoising $a^T \sim \mathcal{N}(0, I) \rightarrow a^0$ using $\Psi_\phi$: $a_t \sim \pi_\phi(x_t)$
    Step environment: $x_{t+1}, r_{t+1} \leftarrow \text{env}(a_t)$
    Store $(x_t, a_t, r_{t+1}, x_{t+1})$ in $\mathcal{B}$
    Sample minibatch of $N$ transitions $(x_t, a_t, r_{t+1}, x_{t+1})$ from $\mathcal{B}$
    Sample actions via iterative denoising using $\Psi_\phi$: $\tilde{a}_{t+1} \sim \pi_\phi(x_{t+1})$
    Compute critic target: $y_t = r_{t+1} + \gamma \min_{i=1,2} Q_{\theta_i'}(x_{t+1}, \tilde{a}_{t+1})$
    Update critics: $\theta_i = \arg\min_{\theta_i} N^{-1} \sum (y_t - Q_{\theta_i}(s_t, a_t))^2$
    Update score model: $\phi = \arg\min_\phi N^{-1} \sum (\Psi_\phi(x_t, a_t) - \nabla_a Q(x_t, a_t))^2$
    Update target networks: $\theta_i' \leftarrow \tau\theta_i + (1 - \tau)\theta_i'$

---

In fig. 3, we evaluate QSM on six tasks from the Deepmind control suite of tasks. These tasks range from easy to medium difficulty. We also compare QSM to SAC (Haarnoja et al., 2018) and TD3 (Fujimoto et al., 2018), and find that QSM manages to achieve similar or better performance than the SAC and TD3 baselines. Conventional actor-critic methods parameterize the actor as a Gaussian distribution or a transformed Gaussian distribution, which can lead to sub-optimal exploration strategies, as certain action modes may get dropped or low Q-value actions may get sampled frequently. QSM instead learns policies which are not constrained to a fixed distribution class and are able to cover many different action modes. As such, QSM learn policies which are multi-model and complex, thereby enabling policies to better approximate the true optimal policy. In fig. 4, we visualize actions sampled from a QSM policy for the initial state of a toy cartpole swingup task. Note the high mass around the extreme actions, both of which represent the initial action for the two unique optimal trajectories.

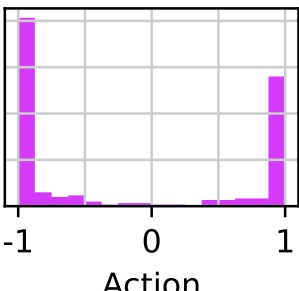

Figure 4: QSM can learn multi-modal policies. Samples from policy shown for the first state of a toy cartpole swingup task, where -1 and 1 represent the initial action for each of two optimal trajectories.

## 6   Conclusion

Diffusion models offer a promising class of models to represent policy distributions, not only because of their expressibility and ease of sampling, but the ability to model the distribution of a policy through its score. To train such diffusion model policies in the reinforcement learning setting, we introduce the Q-score matching (QSM) algorithm, which iteratively matches the parameterized score of the policy to the action gradient of its Q-function. This gives a more geometric viewpoint on how to optimize such policies, namely through iteratively matching vector fields to each other. We additionally provide a practical implementation of this algorithm, and find favorable results when compared to popular RL algorithms.

There are still plenty of avenues for future work. This work focused purely on the score $\Psi$, or the "drift" term of the diffusion model, and assumed a fixed noise model $\Sigma_a$. However, optimization of $\Sigma_a$ is an especially interesting direction for future work; delving deeper into this optimization might reveal intriguing connections to maximum entropy reinforcement learning, providing a richer understanding of the balance between exploration and exploitation. Additionally the theory for this paper builds a baseline for QSM and related algorithms, but there still remain many important theoretical questions to answer. In particular, convergence rate analysis could highlight further improvements to the QSM algorithm.

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

## A  ADDITIONAL EXPERIMENTS

We provide here experiments of QSM on more difficult environments (here Ant/Quadruped and Humanoid, both walk tasks). To push the performance of QSM further, we only added one hyperparameter described in Section 4.3: a scaling parameter for the Q action gradient $\nabla_a Q$ being matched to: that is, the training loss for $\Psi$ is $\|\Psi - \alpha \nabla_a Q\|_2^2$ for some fixed scalar $\alpha > 0$.

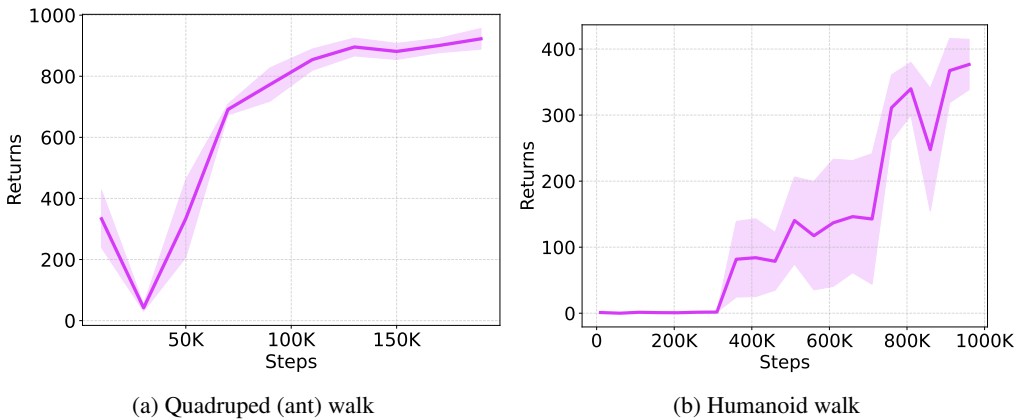

(a) Quadruped (ant) walk          (b) Humanoid walk

Figure 5: Evaluation returns of QSM on more difficult continuous RL tasks, demonstrating potential of similar methods that exploit diffusion model structure. As tasks become more dependent on long-term behavior, the scaling term $\alpha$ needs to be increased further to allow more aggressive exploration further from rewards.

## B  PROOFS

The following proofs build upon each other; for readability, please read all proofs in sequence. Note the following formalities:

1. Important notation and definitions used for this paper (and likewise for the following proofs) are provided in Section 2.

2. The below proofs consider optimizations over smooth collections of vector fields $\Psi : \mathbb{R}^s \times \mathbb{R}^a \to \mathbb{R}^a$, that are bounded in highest norm ($\mathcal{C}^0$ norm of $(s, a) \to \|\Psi(s, a)\|_2$) and are Lipschitz with some Lipschitz constant. This choice was made to give some control over the vector field space (bounded but not closed), but remain flexible enough to allow our local perturbations of vector fields towards optima. It remains an interesting line for future work to tighten the optimization space for the vector fields (e.g. fixed Lipschitz constant as well), alongside a sharper ascent rule (akin to equation 38). For this paper, we wanted to simplify the optimization setting in order to focus theoretical insight on the relation of $\Psi$ to its Q-function action gradient $\nabla_a Q^\Psi$.

3. All norms $\| \cdot \|$, unless specified, are the $L^2$ norm.

**A note for readability.** By far the longest and densest segment is the first part of the proof for Theorem 1, namely the section "aligning at the origin". This is where the majority of our analysis of the Q-function is built, and the rest of the theory will pretty straightforwardly follow from what is built up there. If you understand this portion, the rest of this paper's theory will follow.

### B.1 PROOF OF POLICY GRADIENT FORMULA IN EQUATION 9

We introduce here a formal theorem and proof pair to justify equation 9.

**Theorem 3.** *Suppose we parameterize a policy $\pi_\theta(a|s)$ through an incremental sequence of steps, such that we only have access to marginal probabilities $\pi_\theta(a^\tau|a^{\tau-1}, s)$ that satisfy the following identity:*

$$\pi(a^K|s) = \int_{a^{K-1}} \cdots \int_{a^1} \pi(a^1) \prod_{\tau=2}^{K} \pi(a^\tau|a^{\tau-1}, s) da^1 \cdots da^{K-1}. \tag{22}$$

*The following expectations for the policy gradient are then equal:*

$$\mathbb{E}_{(s_0, a_0, s_1, \ldots)} \sum_{t=1}^{\infty} Q(s_t, a_t) \nabla_\theta \log \pi_\theta(a_t|s_t) \tag{23}$$

$$= \mathbb{E}_{(s_0, \{a_0^\tau\}_{\tau=1}^K, s_1, \ldots)} \sum_{t=1}^{\infty} Q(s_t, a_t^K) \left( \sum_{\tau=1}^{K} \nabla_\theta \log \pi_\theta(a_t^\tau|a_t^{\tau-1}, s_t) \right),$$

*where we denote $\pi_\theta(a_t^1|a_t^0, s) = \pi_\theta(a_t^1|s)$.*

*Proof.* We directly plug in equation 8 in the simplified terms to the LHS to get the following:

$$\mathbb{E}_{(s_0, a_0^K, s_1, \ldots)} \sum_{t=1}^{\infty} Q(s_t, a_t) \nabla_\theta \log \left( \int_{a_t^{K-1}} \cdots \int_{a_t^1} \prod_{\tau=1}^{K} \pi(a_t^\tau|a_t^{\tau-1}, s_t) da^1 \cdots da^{K-1} \right).$$

While logs of sums/integrals do not expand, we can utilize the fact that this is the gradient of a log to re-expand and simplify to the following:

$$(LHS) = \mathbb{E}_{(s_0, a_0^K, s_1, \ldots)} \sum_{t=1}^{\infty} Q(s_t, a_t^K) \frac{1}{\int_{a_t^{K-1}} \cdots \int_{a_t^1} \prod_{\tau=1}^{K} \pi(a_t^\tau|a_t^{\tau-1}, s_t)} \tag{24}$$

$$\cdot \nabla_\theta \int_{a_t^{K-1}} \cdots \int_{a_t^1} \prod_{\tau=1}^{K} \pi(a_t^\tau|a_t^{\tau-1}, s_t) da^1 \cdots da^{K-1},$$

$$= \mathbb{E}_{(s_0, a_0^K, s_1, \ldots)} \sum_{t=1}^{\infty} Q(s_t, a_t^K) \frac{1}{\pi_\theta(a_t^K|s_t)} \int_{a_t^{K-1}} \cdots \int_{a_t^1} \nabla_\theta \prod_{\tau=1}^{K} \pi(a_t^\tau|a_t^{\tau-1}, s_t) da^1 \cdots da^{K-1},$$
$$\tag{25}$$

$$= \mathbb{E}_{(s_0, a_0^K, s_1, \ldots)} \sum_{t=1}^{\infty} \int_{a_t^{K-1}} \cdots \int_{a_t^1} Q(s_t, a_t^K) \frac{1}{\pi_\theta(a_t|s_t)} \left( \prod_{\tau=1}^{K} \pi(a_t^\tau|a_t^{\tau-1}, s_t) \right) \tag{26}$$

$$\cdot \nabla_\theta \log(\prod_{\tau=1}^{K} \pi(a_t^\tau|a_t^{\tau-1}, s_t)) da^1 \cdots da^{K-1},$$

$$= \mathbb{E}_{(s_0, s_1, s_2, \ldots)} \sum_{t=1}^{\infty} \int_{a_t^K} \int_{a_t^{K-1}} \cdots \int_{a_t^1} \left( \prod_{\tau=1}^{K} \pi(a_t^\tau|a_t^{\tau-1}, s_t) \right) \tag{27}$$

$$\cdot Q(s_t, a_t^K) \nabla_\theta \sum_{\tau=1}^{K} \log(\pi(a_t^\tau|a_t^{\tau-1}, s_t)) da^1 \cdots da^K,$$

$$= \mathbb{E}_{(s_0, \{a_0^\tau\}_{\tau=1}^K, s_1, \ldots)} \sum_{t=1}^{\infty} Q(s_t, a_t^K) \left( \sum_{\tau=1}^{K} \nabla_\theta \log(\pi(a_t^\tau|a_t^{\tau-1}, s_t)) \right). \tag{28}$$

$\square$

## B.2  PROOF OF THEOREM 1

*Proof.* We will use the shorthand $Q := Q^{\Psi}$. Note that for any $\tau > 0$, we can decompose $Q(\mathbf{0}, \mathbf{0})$ recursively into the following:

$$Q(\mathbf{0}, \mathbf{0}) = \int_0^{\infty} \gamma^t r(s(t)) dt, \tag{29}$$

$$= \int_0^{\tau} \gamma^t r(s(t)) dt + \int_{\tau}^{\infty} \gamma^t r(s(t)) dt, \tag{30}$$

$$= \int_0^{\tau} \gamma^t r(s(t)) dt + \int_0^{\infty} \gamma^{t+\tau} r(s(t+\tau)) dt, \tag{31}$$

$$= \int_0^{\tau} \gamma^t r(s(t)) dt + \gamma^{\tau} \int_0^{\infty} \gamma^t r(s(t, s(\tau), a(\tau))) dt, \tag{32}$$

$$= \int_0^{\tau} \gamma^t r(s(t)) dt + \gamma^{\tau} Q(s(\tau), a(\tau)). \tag{33}$$

This will form the baseline of all of our local analysis, namely the fact that the Q-function can be broken down into two components:

1. a purely state-dependent integral that can be made arbitrarily small, and

2. a boundary term of the same Q-function.

We proceed with this proof with the contrapositive statement: suppose that, for some $t \in [0, \infty)$, it follows that $\Psi(s(t), a(t))$ is not aligned with $\nabla_a Q(s(t), a(t))$. We show that there exists a "bump" of $\Psi$ towards $\nabla_a Q$ that strictly increases $Q(\mathbf{0}, \mathbf{0})$.

**Aligning at the origin.** Suppose the above contrapositive statement yields $t = 0$, and assume that $\nabla_a Q(\mathbf{0}, \mathbf{0}) \neq \mathbf{0}$, and that $\Psi(\mathbf{0}, \mathbf{0})$ is not collinear with $\nabla_a Q(\mathbf{0}, \mathbf{0})$. We will use this to construct a modified vector field $\Psi'$ such that $Q^{\Psi'}(\mathbf{0}, \mathbf{0}) > Q^{\Psi}(\mathbf{0}, \mathbf{0})$. Suppose we have chosen $\tau$ and $\epsilon$ small enough such that the following all hold:

1. $s(\tau') = \tau' F(\mathbf{0}, \mathbf{0}) + o(\tau')$ for all $0 < \tau' \leq \tau$,

2. $a(\tau') = \tau' \Psi(\mathbf{0}, \mathbf{0}) + o(\tau')$ for all $0 < \tau' \leq \tau$,

3. $r(s) = r(\mathbf{0}) + \langle \nabla_s r(\mathbf{0}), s \rangle + o(\|s\|)$ for all $s, \|s\| \leq \epsilon$,

4. $Q(s, a) = Q(\mathbf{0}, \mathbf{0}) + \langle \nabla_s Q(\mathbf{0}, \mathbf{0}), s \rangle + \langle \nabla_a Q(\mathbf{0}, \mathbf{0}), a \rangle + o(\|s : a\|)$ for all $s, a$ such that $\|s\|, \|a\| \leq \epsilon$, where $(s : a) \in \mathbb{R}^{s \times a}$ is the stacked vector of $s$ and $a$,

5. $\|s(\tau')\| \leq \epsilon, \|a(\tau')\| \leq \epsilon$ for all $0 < \tau' \leq \tau$.

We can then make the following approximations for $Q(\mathbf{0}, \mathbf{0})$:

$$Q(\mathbf{0}, \mathbf{0}) = \int_0^\tau \gamma^t r(s(t))dt + \gamma^\tau Q(s(\tau), a(\tau)), \tag{34}$$

$$= \int_0^\tau \gamma^t \left( r(\mathbf{0}) + \langle \nabla_s r(\mathbf{0}), s(t) \rangle + o(\|s(t)\|) \right) dt \tag{35}$$
$$+ \gamma^\tau \left( Q(\mathbf{0}, \mathbf{0}) + \langle \nabla_s Q(\mathbf{0}, \mathbf{0}), s(\tau) \rangle + \langle \nabla_a Q(\mathbf{0}, \mathbf{0}), a(\tau) \rangle + o(\|s(\tau) : a(\tau)\|) \right),$$

$$= \int_0^\tau \gamma^t \left( r(\mathbf{0}) + t \langle \nabla_s r(\mathbf{0}), F(\mathbf{0}, \mathbf{0}) \rangle + o(t)(\|F(\mathbf{0}, \mathbf{0})\| + \|\nabla_s r(\mathbf{0})\|) \right) dt \tag{36}$$
$$+ \gamma^\tau (Q(\mathbf{0}, \mathbf{0}) + \tau \langle \nabla_s Q(\mathbf{0}, \mathbf{0}), F(\mathbf{0}, \mathbf{0}) \rangle + \tau \langle \nabla_a Q(\mathbf{0}, \mathbf{0}), \Psi(\mathbf{0}, \mathbf{0}) \rangle$$
$$+ o(\tau)(\|\nabla_s Q(\mathbf{0}, \mathbf{0})\| + \|\nabla_a Q(\mathbf{0}, \mathbf{0})\| + \|F(\mathbf{0}, \mathbf{0}) : \Psi(\mathbf{0}, \mathbf{0})\|)),$$

$$= \int_0^\tau \gamma^t \left( r(\mathbf{0}) + t \langle \nabla_s r(\mathbf{0}), F(\mathbf{0}, \mathbf{0}) \rangle \right) dt \tag{37}$$
$$+ \gamma^\tau (Q(\mathbf{0}, \mathbf{0}) + \tau \langle \nabla_s Q(\mathbf{0}, \mathbf{0}), F(\mathbf{0}, \mathbf{0}) \rangle + \tau \langle \nabla_a Q(\mathbf{0}, \mathbf{0}), \Psi(\mathbf{0}, \mathbf{0}) \rangle)$$
$$+ o(\tau).$$

Note the three summands in equation 37 that $Q(\mathbf{0}, \mathbf{0})$ now splits into:

1. An integral term purely dependent on the origin state dynamics $F(\mathbf{0}, \mathbf{0})$,

2. A term with linearly separable dependence on the origin's state dynamics $\Psi(\mathbf{0}, \mathbf{0})$, and

3. A term decaying strictly faster than our discretization term $\tau$.

We then focus our attention on the second summand, as it at least appears to contain the separated dependence of the score $\Psi$. Note, however, that the Q-function itself still has dependence on $\Psi$, so we need to be careful. To finalize the separation of the score, we split into two cases:

1. Suppose there exists some $\epsilon > 0$ such that $a(t)$ will not re-cross the $\epsilon$-ball around the origin, denoted $B_\epsilon$, after its first escape, from any initial condition within $B_\epsilon$. Thus, for this further reduced neighborhood $\epsilon$ (and likewise reduced $\tau$), $Q(\mathbf{0}, \mathbf{0})$ is completely determined by the local integral $\int_0^\tau \gamma^t r(s(t))dt$ while $(s(t), a(t))$ traverses through $B_\epsilon$, and the boundary condition $\gamma^\tau Q(s(\tau), a(\tau))$ once it hits the $\epsilon$-ball boundary. This gives us a neighborhood to manipulate $\Psi$: if the changes to $\Psi$ only happen within $B_\epsilon$, then the value of $Q(s, a)$ on the border of $B_\epsilon$ remains the same.

   Let $\phi : \mathbb{R}^a \to \mathbb{R}$ be a partition of unity with respect to $B_\epsilon$: $\phi$ is a function purely of the norm $\|a\|$, $\phi(a) = 1$ for all $\|a\| \le \frac{\epsilon}{2}$, and $\phi(a) = 0$ for all $\|a\| \ge \epsilon$. Consider the following modified score function:

$$\Psi'(s, a) := (1 - \phi(a))\Psi(s, a) + \phi(a)\frac{\|\Psi(\mathbf{0}, \mathbf{0})\|}{\|\nabla_a Q(\mathbf{0}, \mathbf{0})\|}\nabla_a Q(\mathbf{0}, \mathbf{0}). \tag{38}$$

   By construction, $\|\Psi\|_{\mathcal{C}^0} := \sup_{s,a} \|\Psi(s, a)\| \le C \implies \|\Psi'\|_{\mathcal{C}^0} \le C$, and $\Psi'$ is Lipschitz (of a likely higher but finite constant) as it is the sum and product of Lipschitz functions. Note that if $\Psi(\mathbf{0}, \mathbf{0}) = \mathbf{0}$, then $\Psi' = (1 - \phi(a))\Psi$, in which case we replace $\|\Psi(\mathbf{0}, \mathbf{0})\|$ in equation 38 with an appropriate constant such that $\|\Psi'\|_{\mathcal{C}^0} \le C$.

   We then get the following improvement of $Q(\mathbf{0}, \mathbf{0})$, working around the fact that the Taylor approximation bounds with respect to $\epsilon, \tau$ do not necessarily apply now for the action

component of $\Psi'$:

$$Q^{\Psi'}(\mathbf{0}, \mathbf{0}) - Q^{\Psi}(\mathbf{0}, \mathbf{0}) = \gamma^{\tau} \langle \nabla_a Q(\mathbf{0}, \mathbf{0}), a_{\Psi'}(\tau) - a_{\Psi}(\tau) \rangle + o(\tau), \tag{39}$$

$$= \gamma^{\tau} \left\langle \nabla_a Q(\mathbf{0}, \mathbf{0}), \int_0^{\tau} \Psi'(\mathbf{0}, a(\tau)) dt - \tau \Psi(\mathbf{0}, \mathbf{0}) \right\rangle$$
$$+ o(\tau), \tag{40}$$

$$= \gamma^{\tau} \left\langle \nabla_a Q(\mathbf{0}, \mathbf{0}), \int_0^{\tau} \Psi'(\mathbf{0}, a(t)) dt - \int_0^{\tau} \Psi(\mathbf{0}, \mathbf{0}) dt \right\rangle$$
$$+ o(\tau), \tag{41}$$

$$= \gamma^{\tau} \int_0^{\tau} \langle \nabla_a Q(\mathbf{0}, \mathbf{0}), \Psi'(\mathbf{0}, a(t)) - \Psi(\mathbf{0}, \mathbf{0}) \rangle \, dt$$
$$+ o(\tau), \tag{42}$$

$$= \gamma^{\tau} \ell(\tau) + o(\tau), \tag{43}$$

where $\ell(\tau) := \int_0^{\tau} \langle \nabla_a Q(\mathbf{0}, \mathbf{0}), \Psi'(\mathbf{0}, a(\tau)) - \Psi(\mathbf{0}, \mathbf{0}) \rangle \, dt$. Note that the desired inequality follows if $\ell(\tau) \geq M\tau$ for some $M > 0$, since this would imply $\lim_{\tau \to 0} \frac{o(\tau)}{\gamma^{\tau} \ell(\tau)} = 0$ and further imply there exists some $\tau$ such that $|\gamma^{\tau} \ell(\tau)| > |o(\tau)|$, and thus further that $Q^{\Psi'}(\mathbf{0}, \mathbf{0}) > Q^{\Psi}(\mathbf{0}, \mathbf{0})$.

It then remains to find a constant $M$ such that $\ell(\tau) > M\tau$. Split the integral at hand as follows:

$$\int_0^{\tau} \langle \nabla_a Q(\mathbf{0}, \mathbf{0}), \Psi'(\mathbf{0}, a(\tau)) - \Psi(\mathbf{0}, \mathbf{0}) \rangle \, dt \tag{44}$$

$$= \int_0^{\tau_{\epsilon/2}} \langle \nabla_a Q(\mathbf{0}, \mathbf{0}), \Psi'(\mathbf{0}, a(\tau)) - \Psi(\mathbf{0}, \mathbf{0}) \rangle \, dt$$
$$+ \int_{\tau_{\epsilon/2}}^{\tau} \langle \nabla_a Q(\mathbf{0}, \mathbf{0}), \Psi'(\mathbf{0}, a(\tau)) - \Psi(\mathbf{0}, \mathbf{0}) \rangle \, dt,$$

$$= \tau_{\epsilon/2} \left\langle \nabla_a Q(\mathbf{0}, \mathbf{0}), \frac{\|\Psi(\mathbf{0}, \mathbf{0})\|}{\|\nabla_a Q(\mathbf{0}, \mathbf{0})\|} \nabla_a Q(\mathbf{0}, \mathbf{0}) \right\rangle \tag{45}$$

$$+ \int_{\tau_{\epsilon/2}}^{\tau} \langle \nabla_a Q(\mathbf{0}, \mathbf{0}), \Psi'(\mathbf{0}, a(\tau)) - \Psi(\mathbf{0}, \mathbf{0}) \rangle \, dt,$$

$$\geq \tau_{\epsilon/2} M' \tag{46}$$

where $\tau_{\epsilon/2}$ is the hitting time of $a(t)$ at $\|a(t)\| = \epsilon/2$, $M' := \left\langle \nabla_a Q(\mathbf{0}, \mathbf{0}), \frac{\|\Psi(\mathbf{0}, \mathbf{0})\|}{\|\nabla_a Q(\mathbf{0}, \mathbf{0})\|} \nabla_a Q(\mathbf{0}, \mathbf{0}) \right\rangle > 0$, and the second integral is non-negative by non-collinearity of $\nabla_a Q(\mathbf{0}, \mathbf{0})$ and $\Psi(\mathbf{0}, \mathbf{0})$, along with the construction of $\Psi'$ as convex combinations of these two vectors. Finally, we can further find some $M > 0$ such that $\tau_{\epsilon/2} M' \geq M\tau$, since the flow of the original $a(t)$ is governed by a constant vector up to error $o(\tau)$, and thus the constructed $\epsilon$ scales at fastest linearly towards $\epsilon \to 0$ (and likewise for the modified $a_{\Psi'}(t)$ within $\epsilon/2$), noting that $\epsilon$ can be chosen separately and arbitrarily for the action dynamics if $\Psi(\mathbf{0}, \mathbf{0}) = \mathbf{0}$. This then concludes that $\ell(\tau) \geq M\tau$ for some $M > 0$, and concludes this part of the proof.

2. Suppose however we cannot find such a further reduced $\epsilon$ (this can happen for example in certain attractor systems). We then leave $\epsilon$ as-is up to this point, and similarly construct $\Psi'$ from equation 38. Let $\mathcal{T} := \{T_1, T_1^e, T_2, T_2^e, \ldots\}$ be a monotonically increasing set of positive times with respect to the modified score $\Psi'$ such that $a_{\Psi'}(t)$ crosses into $B_{\epsilon}$ at each time $T_i$ and subsequently back out at time $T_i^e$ ($T_1$ is the first re-entry of $a(t)$ *after* it first leaves from $t = 0$). Since $\nabla_a Q(\mathbf{0}, \mathbf{0}) \neq 0$, the origin is not an equilibrium, and each $T_i$ has a paired $T_i^e$.

**If $\mathcal{T}$ is empty** (note this is technically still possible, as there can still exist *some* initial condition along the boundary of $B_{\epsilon}$ that does not re-enter), we repeat the analysis above from the existence of a non-re-enterable epsilon-ball $B_{\epsilon}$.

**If $\mathcal{T}$ is finite**, we can make the following decomposition of $Q^{\Psi'}(\mathbf{0}, \mathbf{0})$ for some $k \in \mathbb{N}$:

$$Q^{\Psi'}(\mathbf{0}, \mathbf{0}) = \int_0^{T_1} \gamma^t r(s_{\Psi'}(t))dt + \int_{T_1}^{T_1^e} \gamma^t r(s_{\Psi'}(t))dt \tag{47}$$

$$+ \int_{T_1^e}^{T_2} \gamma^t r(s_{\Psi'}(t))dt + \int_{T_2}^{T_2^e} \gamma^t r(s_{\Psi'}(t))dt$$

$$+ \ldots$$

$$+ \int_{T_{k-1}^e}^{T_k} \gamma^t r(s_{\Psi'}(t))dt + \int_{T_k}^{T_k^e} \gamma^t r(s_{\Psi'}(t))dt + \int_{T_k^e}^{\infty} \gamma^t r(s_{\Psi'}(t))dt,$$

$$= \int_0^{T_1} \gamma^t r(s_{\Psi'}(t))dt + \int_{T_1}^{T_1^e} \gamma^t r(s_{\Psi'}(t))dt \tag{48}$$

$$+ \ldots$$

$$+ \int_{T_{k-1}^e}^{T_k} \gamma^t r(s_{\Psi'}(t))dt + \int_{T_k}^{T_k^e} \gamma^t r(s_{\Psi'}(t))dt$$

$$+ \gamma^{T_k^e} Q^{\Psi'}(s_{\Psi'}(T_k^e), a_{\Psi'}(T_k^e)),$$

Focusing now on the last two summands of equation 48, we can make the following simplifications, with explanations below:

$$\int_{T_{k-1}^e}^{T_k} \gamma^t r(s_{\Psi'}(t))dt + \int_{T_k}^{T_k^e} \gamma^t r(s_{\Psi'}(t))dt + \gamma^{T_k^e} Q^{\Psi'}(s_{\Psi'}(T_k^e), a_{\Psi'}(T_k^e)), \tag{49}$$

$$= \int_{T_{k-1}^e}^{T_k} \gamma^t r(s_{\Psi'}(t))dt + \int_{T_k}^{T_k^e} \gamma^t r(s_{\Psi'}(t))dt + \gamma^{T_k^e} Q^{\Psi}(s_{\Psi'}(T_k^e), a_{\Psi'}(T_k^e)), \tag{50}$$

$$> \int_{T_{k-1}^e}^{T_k} \gamma^t r(s_{\Psi'}(t))dt + \int_{T_k}^{T_k^e} \gamma^t r(s_{\Psi'}(t))dt + \gamma^{T_k^e} Q^{\Psi}(s_{\Psi'_{T_k}}(T_k^e), a_{\Psi'_{T_k}}(T_k^e)), \tag{51}$$

$$= \int_{T_{k-1}^e}^{T_k} \gamma^t r(s_{\Psi'}(t))dt \tag{52}$$

$$+ \gamma^{T_k} \left( \int_0^{T_k^e - T_k} \gamma^t r(s_{\Psi'_{T_k}}(t))dt + o(\tau) + \gamma^{T_k^e - T_k} Q^{\Psi}(s_{\Psi'_{T_k}}(T_k^e), a_{\Psi'_{T_k}}(T_k^e)) \right),$$

$$= \int_{T_{k-1}^e}^{T_k} \gamma^t r(s_{\Psi'}(t))dt + \gamma^{T_k} Q^{\Psi}(s_{\Psi'}(T_k), a_{\Psi'}(T_k)) + \gamma^{T_k} o(\tau), \tag{53}$$

$$= \int_{T_{k-1}^e}^{T_k} \gamma^t r(s_{\Psi'_{T_{k-1}^e}}(t))dt + \gamma^{T_k} Q^{\Psi}(s_{\Psi'}(T_k), a_{\Psi'}(T_k)) + \gamma^{T_k} o(\tau), \tag{54}$$

$$= \gamma^{T_{k-1}^e} Q^{\Psi}(s_{\Psi'}(T_{k-1}^e), a_{\Psi'}(T_{k-1}^e)) + \gamma^{T_k} o(\tau), \tag{55}$$

where $s_{\Psi'_T}(t)$ for $T < t$ is the resulting state $s(t)$ after integrating using $\Psi'$ until $T$, then switching to $\Psi$.

(a) Equation 50 follows by definition of $T_k^e$ being the last time the system $a(t)$ crosses $B_\epsilon$, so $\Psi = \Psi'$ for the rest of the path and $Q^{\Psi'}(s_{\Psi'}(T_k^e), a_{\Psi'}(T_k^e)) = Q^{\Psi}(s_{\Psi'}(T_k^e), a_{\Psi'}(T_k^e))$.

(b) Equation 51 follows from the same analysis in part 1, since we can now fix our analysis on the originally Taylor-approximated $Q^{\Psi}$. Note that we now need to integrate $\Psi'$ through the whole neighborhood $B_\epsilon$ rather than just the sub-neighborhood $B_{\epsilon/2}$ where $\Psi'$ is constant, but the analysis remains the same, noting the linearity at equation 35 even before taking an approximation of the dynamical system's flow.

(c) Equation 52 comes from a reparamaterization of integrals, and using the analysis for equation 37 to get an approximation of the internal reward integral $\int_{T_k}^{T_k^e} \gamma^t r(s_{\Psi'}(t))dt$ that is unchanged with respect to the action dynamics $\Psi$ and $\Psi'$, with error $o(\tau)$. Note $\tau$ here is the max difference between $T_i$ and $T_i^e$; this can be arbitrarily decreased with $\epsilon$ since $\Psi$ is Lipschitz.

(d) Equation 53 comes from wrapping the recursive equation for the Q-function (see equation 33), noting we have now removed dependence on $\Psi'$ from time $T_k$ onwards.

(e) Equation 54 comes from the fact that, by construction of $\mathcal{T}$, $\Psi' = \Psi$ on the time interval $[T_{k-1}^e, T_k]$.

(f) Finally, equation 55 is another wrapping of the recursive equation for the Q-function.

The proof strategy now becomes clear: we can iteratively repeat the process of equation 50 through equation 55 to eventually write the RHS in terms of $Q^\Psi$ and a rapidly decaying $o(\tau)$ term, achieving the desired inequality.

Repeat the above process inductively, where we conclude with the following:

$$Q^{\Psi'}(\mathbf{0}, \mathbf{0}) > \int_0^{T_1} \gamma^t r(s_{\Psi'}(t))dt + \int_{T_1}^{T_1^e} \gamma^t r(s_{\Psi'}(t))dt \tag{56}$$

$$+ \gamma^{T_1^e} Q^\Psi(s_{\Psi'}(T_1^e), a_{\Psi'}(T_1^e)) + (\sum_{i=2}^k \gamma^{T_i})o(\tau),$$

$$> \int_0^{T_1} \gamma^t r(s_{\Psi'}(t))dt + \int_{T_1}^{T_1^e} \gamma^t r(s_{\Psi'}(t))dt \tag{57}$$

$$+ \gamma^{T_1^e} Q^\Psi(s_{\Psi'_{T_1}}(T_1^e), a_{\Psi'_{T_1}}(T_1^e)) + (\sum_{i=2}^k \gamma^{T_i})o(\tau),$$

$$= \int_0^\tau \gamma^t r(s_{\Psi'}(t))dt + \int_\tau^{T_1} \gamma^t r(s_{\Psi'}(t))dt \tag{58}$$

$$+ \gamma^{T_1} Q^\Psi(s_{\Psi'}(T_1), a_{\Psi'}(T_1)) + (\sum_{i=1}^k \gamma^{T_i})o(\tau),$$

$$= \int_0^\tau \gamma^t r(s_{\Psi'}(t))dt + \gamma^\tau Q^\Psi(s_{\Psi'}(\tau), a_{\Psi'}(\tau)) + (\sum_{i=1}^k \gamma^{T_i})o(\tau), \tag{59}$$

$$= Q^{\Psi'}(\mathbf{0}, \mathbf{0}) + \gamma^\tau \ell(\tau) + (1 + \sum_{i=1}^k \gamma^{T_i})o(\tau). \tag{60}$$

Since $\lim_{\tau \to 0} \frac{(1 + \sum_{i=1}^k \gamma^{T_k})o(\tau)}{\gamma^\tau \ell(\tau)} = 0$, there exists some $\tau$ such that the above equation implies $Q^{\Psi'}(\mathbf{0}, \mathbf{0}) > Q^\Psi(\mathbf{0}, \mathbf{0})$.

**If $\mathcal{T}$ is infinite**, we no longer have our "base case" for the above induction. However, by using the decaying structure of the Q-function, we can artificially create a base case. Noting that $0 \leq Q^\Psi(s, a), Q^{\Psi'}(s, a) \leq \int_0^\infty \gamma^t dt = (\log(\gamma^{-1}))^{-1}$, we can write the following for any fixed $k$ for some error term $q_k$:

$$\gamma^{T_k^e} Q^{\Psi'}(s_{\Psi'}(T_k^e), a_{\Psi'}(T_k^e)) = \gamma^{T_k^e} Q^\Psi(s_{\Psi'}(T_k^e), a_{\Psi'}(T_k^e)) + \gamma^{T_k^e} q_k, \tag{61}$$

where each $|q_k| \leq (\log(\gamma^{-1}))^{-1}$. Thus, we can repeat the analysis from the finite $\mathcal{T}$ case to get the following:

$$Q^{\Psi'}(\mathbf{0}, \mathbf{0}) > Q^\Psi(\mathbf{0}, \mathbf{0}) + \gamma^\tau \ell(\tau) + (1 + \sum_{i=1}^k \gamma^{T_i})o(\tau) + \gamma^{T_k^e} q_k. \tag{62}$$

Since $\sum_{i=1}^\infty \gamma^{T_i} < \infty$, it follows that $\lim_{\tau \to 0} \frac{(1 + \sum_{i=1}^\infty \gamma^{T_i})o(\tau)}{\gamma^\tau \ell(\tau)} = 0$, and there exists some $\tau$ and some constant $\kappa$ such that $\gamma^\tau \ell(\tau) + (1 + \sum_{i=1}^k \gamma^{T_i})o(\tau) > \kappa > 0$, regardless of

what value of $k$ is chosen. Since $\mathcal{T}$ is infinite and $q_k$ is bounded, we can choose some $k'$ such that $|\gamma^{T^e_{k'}} q_{k'}| < \kappa$. Under such chosen $k'$, it follows that $Q^{\Psi'}(\mathbf{0}, \mathbf{0}) > Q^\Psi(\mathbf{0}, \mathbf{0})$.

**Aligning beyond the origin.** Suppose that $\Psi$ is aligned with $\nabla_a Q$ until some positive $t' > 0$. Split $Q(\mathbf{0}, \mathbf{0}) = \int_0^{t'-\tau} r(s(t))dt + \gamma^{t'-\tau} \left( \int_0^\tau \gamma^t r(s(t + t' - \tau))dt + \gamma^\tau Q(s(t), a(t)) \right)$, and repeat the analysis above on $Q(s(t), a(t))$, noting both that the first summand is unaffected by changing $\Psi$ to $\Psi'$ and that $\tau$ can be chosen small enough for $\int_0^\tau \gamma^t r(s(t + t' - \tau))dt$ to only be dependent on state dynamics, with error $o(\tau)$. $\qquad\square$

**Extensions.** A trivial extension of Theorem 1 is to any starting condition $Q(s_0, a_0)$, and one we can cover briefly is Corollary 1, where we consider expectations over initial conditions of the form $\mathbb{E}_{\mathbb{P} \times \pi} Q^\Psi(s, a)$. Since the analysis for Theorem 1 made perturbations in strictly positive measure regions, we can repeat the above analysis to nudge $\Psi$ at any non-aligned point $(s, a)$ to not only conclude a strict increase in $Q$ at $(s, a)$, but in a measure-positive region around $(s, a)$, and since this change does not decrease $Q(s', a')$ at any other pair $(s', a')$ by repeating the "aligning beyond the origin" analysis, such constructions of updated scores $\Psi'$ in the proof of Theorem 1 similarly conclude Corollary 1.

## B.3 Proof of Theorem 2

*Proof.* Firstly, let's formalize our notion of the support of this full joint distribution with the initial condition distribution and the stochastic dynamics of $s(t), a(t)$:

**Definition 1.** *A point $(s, a) \in \mathbb{R}^s \times \mathbb{R}^a$ is in the support of the fully joint distribution between initial condition distribution $\mathbb{P}, \pi$ and stochastic dynamics of $(s(t), a(t))$ if for every $\epsilon > 0$, the probability that $a(t), s(t) \in B_\epsilon(s, a)$ at some finite time $t$ is strictly greater than 0.*

Using this definition, we then pave a clear path forward. We first repeat the first part analysis from the proof of Theorem 1; namely, we first focus on purely $Q(\mathbf{0}, \mathbf{0})$ and the case where the misalignment happens at the origin, then build up to the theorem with straightforward corollaries.

We then first focus on $Q(\mathbf{0}, \mathbf{0})$ and assume that $\nabla_a Q(\mathbf{0}, \mathbf{0})$ is not collinear with $\Psi(\mathbf{0}, \mathbf{0})$, using this to construct a score $\Psi'$ such that $Q^{\Psi'}(\mathbf{0}, \mathbf{0}) > Q^\Psi(\mathbf{0}, \mathbf{0})$.

We start with our new recursive equation for the Q-function for any $\tau > 0$. Note that any expectations $\mathbb{E}$ are taken with respects to the stochastic dynamics of $s(t), a(t)$.

$$Q(\mathbf{0}, \mathbf{0}) = \mathbb{E} \int_0^\infty \gamma^t r(s(t))dt, \tag{63}$$

$$= \mathbb{E}_{s,a}[\mathbb{E}[\int_0^\infty \gamma^t r(s(t))dt \mid (s(\tau), a(\tau)) = (s, a)]], \tag{64}$$

$$= \mathbb{E}_{s,a}[\mathbb{E}[\int_0^\tau \gamma^t r(s(t))dt + \gamma^\tau Q(s, a) \mid (s(\tau), a(\tau)) = (s, a)]], \tag{65}$$

$$= \mathbb{E}_{s,a}[\mathbb{E}[\int_0^\tau \gamma^t r(s(t))dt \mid (s(\tau), a(\tau)) = (s, a)]]$$
$$+ \gamma^\tau \mathbb{E}_{s,a}[\mathbb{E}[Q(s, a) \mid (s(\tau), a(\tau)) = (s, a)]], \tag{66}$$

$$= \mathbb{E}[\int_0^\tau \gamma^t r(s(t))dt] + \gamma^\tau \mathbb{E}_{s,a}[Q(s, a) \mid (s(\tau), a(\tau)) = (s, a)]. \tag{67}$$

Our new conditions for $\epsilon, \tau$ will be the following:

1. For all $\tau' \leq \tau$, the distribution of $a(\tau')$ is close to $\mathcal{N}(\tau'\Psi(\mathbf{0}, \mathbf{0}), \tau'\Sigma_a(\mathbf{0}, \mathbf{0}))$ in the following sense: for any fixed smooth $f : \mathbb{R}^a \to \mathbb{R}$, the difference between $\mathbb{E}[f(a(\tau'))]$ and $\mathbb{E}_{N(\tau'\Psi(\mathbf{0},\mathbf{0}),\tau')}f(a(\tau'))$ is controlled by some little-$o$ function $o_f(\tau')$: $\mathbb{E}[f(a(\tau'))] - \mathbb{E}_{N(\tau'\Psi(\mathbf{0},\mathbf{0}),\tau'\Sigma_a(\mathbf{0},\mathbf{0}))}[f(a(\tau))] = o_f(\tau')$,

2. $s(\tau')$ is close in distribution to $\mathcal{N}(\tau F(\mathbf{0},\mathbf{0}), \tau \Sigma_s(\mathbf{0},\mathbf{0}))$ with high probability in a similar sense to the above: for any fixed smooth $f : \mathbb{R}^s \to \mathbb{R}$, $\mathbb{E}[f(s(\tau'))] - \mathbb{E}_{N(\tau F(\mathbf{0},\mathbf{0}), \tau \Sigma_s(\mathbf{0},\mathbf{0}))}[f(s(\tau))] = o_f(\tau')$,

3. $r(s) = r(\mathbf{0}) + \langle \nabla_s r(\mathbf{0}), s \rangle + o(\|s\|)$ for all $s, \|s\| \leq \epsilon$.

4. $Q(s,a) = Q(\mathbf{0},\mathbf{0}) + \langle \nabla_s Q(\mathbf{0},\mathbf{0}), s \rangle + \langle \nabla_a Q(\mathbf{0},\mathbf{0}), a \rangle + o(\|s:a\|)$ for all $s, a$ such that $\|s\|, \|a\| \leq \epsilon$.

5. $\tau$ is chosen small enough so that the distribution of paths is essentially similar to restricting the norm of each of $s(\tau), a(\tau)$ to be at most $\epsilon$ in the following sense: for any fixed smooth $f : \mathbb{R}^s \times \mathbb{R}^a \to \mathbb{R}$, it follows that $\mathbb{E}[f(s(\tau), a(\tau))] - \mathbb{E}_{\|s(\tau)\|, \|a(\tau)\| \leq \epsilon}[f(s(\tau), a(\tau))] = o_f(\tau)$.

We now repeat the analysis in equation 35 to equation 37 to get the following:

$$Q(\mathbf{0},\mathbf{0}) = \mathbb{E}\left[\int_0^\tau \gamma^t r(s(t)) dt\right] + \gamma^\tau \mathbb{E}_{s,a}[Q(s,a) \mid (s(\tau), a(\tau)) = (s,a)] \tag{68}$$

$$= \mathbb{E}_{\|s(t)\|, \|a(t)\| \leq \epsilon}\left[\int_0^\tau \gamma^t r(s(t)) dt\right] + \gamma^\tau \mathbb{E}_{\|s\|, \|a\| \leq \epsilon}[Q(s,a) \mid (s(\tau), a(\tau)) = (s,a)]$$
$$\quad + (\log(\gamma^{-1}))^{-1} o(\tau), \tag{69}$$

$$= \mathbb{E}_{\|s(t)\|, \|a(t)\| \leq \epsilon} \int_0^\tau \gamma^t \left(r(\mathbf{0}) + t \langle \nabla_s r(\mathbf{0}), F(\mathbf{0},\mathbf{0}) \rangle\right) dt \tag{70}$$
$$\quad + \gamma^\tau \mathbb{E}_{\|s(\tau)\|, \|a(\tau)\| \leq \epsilon}\left[Q(\mathbf{0},\mathbf{0}) + \langle \nabla_s Q(\mathbf{0},\mathbf{0}), s(\tau) \rangle + \langle \nabla_a Q(\mathbf{0},\mathbf{0}), a(\tau) \rangle\right]$$
$$\quad + o(\tau),$$

$$= \int_0^\tau \gamma^t \left(r(\mathbf{0}) + t \langle \nabla_s r(\mathbf{0}), F(\mathbf{0},\mathbf{0}) \rangle\right) dt \tag{71}$$
$$\quad + \gamma^\tau \mathbb{E}\left[Q(\mathbf{0},\mathbf{0}) + \langle \nabla_s Q(\mathbf{0},\mathbf{0}), s(\tau) \rangle + \langle \nabla_a Q(\mathbf{0},\mathbf{0}), a(\tau) \rangle\right]$$
$$\quad + o(\tau),$$

$$= \int_0^\tau \gamma^t \left(r(\mathbf{0}) + t \langle \nabla_s r(\mathbf{0}), F(\mathbf{0},\mathbf{0}) \rangle\right) dt \tag{72}$$
$$\quad + \gamma^\tau (Q(\mathbf{0},\mathbf{0}) + \tau \langle \nabla_s Q(\mathbf{0},\mathbf{0}), F(\mathbf{0},\mathbf{0}) \rangle + \tau \langle \nabla_a Q(\mathbf{0},\mathbf{0}), \Psi(\mathbf{0},\mathbf{0}) \rangle)$$
$$\quad + \gamma^\tau \sqrt{\tau}(\langle \nabla_s Q(\mathbf{0},\mathbf{0}), \mathbb{E}_{z \sim \mathcal{N}(\mathbf{0}, \Sigma_s(\mathbf{0},\mathbf{0}))} z \rangle + \langle \nabla_a Q(\mathbf{0},\mathbf{0}), \mathbb{E}_{z \sim \mathcal{N}(\mathbf{0}, \Sigma_a(\mathbf{0},\mathbf{0}))} z \rangle) + o(\tau),$$

$$= \int_0^\tau \gamma^t \left(r(\mathbf{0}) + t \langle \nabla_s r(\mathbf{0}), F(\mathbf{0},\mathbf{0}) \rangle\right) dt \tag{73}$$
$$\quad + \gamma^\tau (Q(\mathbf{0},\mathbf{0}) + \tau \langle \nabla_s Q(\mathbf{0},\mathbf{0}), F(\mathbf{0},\mathbf{0}) \rangle + \tau \langle \nabla_a Q(\mathbf{0},\mathbf{0}), \Psi(\mathbf{0},\mathbf{0}) \rangle$$
$$\quad + o(\tau).$$

Since equation 73 is exactly the same as equation 37, we can proceed exactly as we did in the non-stochastic case (proof of Theorem 1), conditioning on events and using the tower rule to exploit specific path structure as we did in the non-stochastic case (e.g. conditioning on the number of times each sampled path $(s(t), a(t))$ re-crosses the $\epsilon$-ball at the origin). Definition 1, along with previous analysis, ensure that any such local perturbations $\Psi'$ result in a strict increase of the Q-function, both at a fixed starting condition and a distribution over starting conditions. $\quad\square$

