# OpenReview forum: "Learning a Diffusion Model Policy from Rewards via Q-Score Matching"
_ICLR.cc/2024/Conference — Submitted to ICLR 2024_

### Official Review · Reviewer_pZv9 · 2023-10-28

**Soundness:** 3 good
**Presentation:** 3 good
**Contribution:** 4 excellent
**Rating:** 5
**Confidence:** 4

**Summary:**

This article proposes learning a diffusion model policy that exploits the linked structure between the score of the policy and the action gradient of the Q-function. The proposed algorithm, called Q-score matching (QSM),  iteratively matches the parameterized score of the policy to the action gradient of its Q-function, giving a more geometric viewpoint on optimizing such policies. The authors provide theoretical justification for the proposed method, revealing the connection between the QSM update rule and the soft policy iteration.

**Strengths:**

1. The authors introduce a novel perspective for policy optimization, which involves learning the score of the policy by utilizing the action gradient of the Q-function.
2. The authors provide rigorous mathematical derivations and theoretical support for the proposed method, establishing its solid foundation.
3. The authors define the decision-making process from the perspective of "score," offering a fresh angle that may inspire further research in this area.

**Weaknesses:**

1. Compared to the rigorous theoretical aspect, the experimental section of the paper appears relatively weak. It only compares the proposed method with two relatively outdated baselines, SAC and TD3, on six simple tasks from the deepmind control suite.
2. In the current experiments, the performance of QSM does not seem to be as impressive. Except for the "Hopper Hop" environment, the baselines seem to be able to outperform QSM in terms of convergence on other environments.
3. QSM is not the first algorithm that utilizes the Q function to guide policy updates. Diffusion-QL proposed by Wang et al. and QGPO proposed by Lu et al. both employ different approaches to leverage the Q function for policy learning. Although QSM is an off-policy online RL algorithm, it would be better to compare it with other methods in an offline setting to evaluate the strengths and weaknesses of different ways to utilize the Q function.

**Questions:**

Can the authors test the performance of the QSM in more challenging environments?

---

> ### Author Response · Authors · 2023-11-16
>
> Thank you for your review and for acknowledging the novelty and potential to inspire future research. We address the points you made a little out of order, as our responses to 1. and 2. build off of our response to point 3.
>
> **3:**    Both referenced papers rely on a behavior cloning term, while the main purpose of this paper is to make a theoretical framework connecting intrinsic structure of a diffusion model policy (here its direct representation of a distribution’s score) to Q-learning. While offline-RL experiments do not fall in this paper’s targeted domain, we do understand the point you make with respect to alternative methods of using the Q-function for diffusion policies. We also discuss sensible alternatives in our responses to Reviewers uSr1 and VG7o. To re-summarize here: a Diffusion-QL-type loss of backpropagating through the computation of Q through sample actions $a$ requires differentiating through a noise model $\phi$ applied on itself $K$ times (for a diffusion model with $K$ denoising steps), while Q-score matching methods only have to differentiate through $\phi$ itself, lending itself more robust and scalable to increases in diffusion model depth, even in the middle of training and evaluation. Policy gradient-type methods avoid the gradient of $\phi^{\circ k}$ problem, but become much less sample efficient as the new expectation (given in eq. (9) of our paper) requires many different samples from the diffusion model of that sampled action to converge to the true policy gradient, a luxury not afforded in online RL.
>
> **1:**    The experiments are not meant to provide a new state-of-the-art model, but to demonstrate validity of this update rule for diffusion model policies in more interpretable online RL environments, which to our knowledge is first demonstrated in literature here. There are plenty of interesting directions to improve on Q-score matching algorithms into a state-of-the-art continuous RL algorithm, but we feel the theory is dense and rich enough to deserve its own paper, and position it to lay the foundation for how to use diffusion model structure to design more efficient diffusion policy training algorithms and spark future research.
>
> **2:**    As discussed with Reviewer VG7o, the main demonstration for these results is not the maximum saturated reward return, but the number of samples it takes QSM to reach a high reward state. The emphasis of this paper is to provide a new theoretical framework that can more fully utilize the structure of diffusion models to write more data-efficient update rules for diffusion model policies.
>
> If you feel any of this discussion should be included in the main body, please let us know. We hope this clarifies your questions and points. If so, please consider modifying your scores and rating to reflect this. And if not, we are of course happy to continue the discussion.

---

### Official Review · Reviewer_VG7o · 2023-11-01

**Soundness:** 2 fair
**Presentation:** 4 excellent
**Contribution:** 3 good
**Rating:** 5
**Confidence:** 3

**Summary:**

This paper introduces a novel reinforcement learning algorithm based on the diffusion model. It matches the score of the policy with the gradient of the Q-function, which better exploits the score-based structure of diffusion models. Both theoretical analysis and empirical evaluation are provided to support the advantage of the proposed method.

**Strengths:**

1. The idea introduced in this paper is quite novel compared with previous works in utilizing the diffusion model in decision-making, which I think may have a profound impact on future works.
2. The paper is well-written. A clear motivation is presented and thorough theoretical analyses are provided to support the model. Figure 1 gives a clear illustration of the effect of Q-score matching.
3. The empirical evaluation shows promising results.

**Weaknesses:**

I feel the experimental part is not well performed by the authors, without much analysis. If the authors can address my concerns below, I'd be happy to raise my score.

First, the authors only visualize the learned action distribution in Figure 4, where a comparison with baselines is missing. SAC is also designed to improve its exploration ability, how is the strategy of QSM compared to that?

Second, QSM only performs the best 3 out of 6 environments, with the comparison to 2 baselines. My suggestion is to add more baselines like policy-gradient methods, and other RL+diffusion models if any. Also, does QSM still gain an advantage in the environment with a discrete action space?

Finally, I'm not sure if the authors have an explanation for the inferior performance of QSM on the Cheetah Run and Walker Run, there seems to exist a clear saturated stage for QSM.

**Questions:**

See questions in the last part.

---

> ### Author Response · Authors · 2023-11-16
>
> Thank you for your review and for acknowledging the novelty and potential for profound future impact. We are quite excited about this work and hope it sparks new directions in both the theory and practice of diffusion model agents.
>
> To address your three questions, we start with a more general comment about the role of this paper and why we wrote it. To the best of our knowledge, all previous work using diffusion model agents was performed in the offline setting, utilizing a pre-existing dataset to train on. In such settings, it's reasonable to use a number of pre-existing methods. However, in online/off-policy RL, data is a much more valuable resource, and we need to make sure we’re exploiting both its structure and the structure of the agent, here a diffusion model, to the best of our ability. To this end, we develop a theoretical framework connecting the intrinsic structure of a diffusion model policy (here its direct representation of a distribution’s score) to Q-learning.
>
> The experiments’ role is to then verify the theoretical framework’s validity in making diffusion model policies data-efficient; here this translates to the quickness of QSM to reach a high-reward state, which it shows more promising results with respect to. There are plenty of interesting directions to improve on QSM into a state-of-the-art continuous RL algorithm, but we feel the theory is dense and rich enough to deserve its own paper, and position it to lay the foundation for how to use diffusion model structure to design more efficient diffusion policy training algorithms and spark future research.
>
> For the questions you raised:
>
> 1. Figure 4 was presented to show that QSM meets some baselines to be an interesting method to explore, rather than claiming novelty in that aspect of our algorithm or improvement over prior methods; here demonstrating that even converged policies still explore alternative paths without any explicit regularization terms. This also addresses the sub-question in question 2 about discrete state spaces: Section 4.3 is solely for pedagogy for the theoretical framework, and practice is solely focused on continuous state/actions spaces where diffusion models are more well-posed.
>
> 2. As discussed above, the main parts of these results that we find interesting is the sample-efficiency of QSM in an online RL setting (how fast the model gets to a high reward state). The online RL setting also proposed some unique challenges for comparable methods. Taking policy gradients for example, seeing eq. (9) in our paper, there is now an explicit dependency on inner-sampled actions in the expectation–if we expect finite sample estimations to converge to the true policy gradient, we need many different “diffusion paths” to the same final sampled action, of which we only get 1 in RL. Standard policy gradients are thus quite sample inefficient for diffusion model policy off-policy/online-RL, which we have empirically verified in the paper’s environments. As the purpose of our experiments is to demonstrate validity in Q-score matching as a learning framework, we did not include these in the paper, but if the reviewer finds these experiments would help strengthen the main claims of this paper, we can include these.
>
> 3. More empirical investigation needs to happen in order to push Q-score learning based methods for competitive performance. The purpose of this paper is to establish a theoretical framework for using the underlying structure of diffusion models (here the score-based structure) in order to design more efficient algorithms for training such policies. Much of this rich theory is developed and presented in the proofs in the appendix; while important for reading, they are quite long and dense, and would unfortunately greatly hinder readability if crammed in the main body.
>
> We hope we have clarified your questions and points. If you feel any of these points should be added or emphasized in the main paper, please let us know. And if any points remain, we are of course happy to continue discussion.

---

### Official Review · Reviewer_uSr1 · 2023-11-04

**Soundness:** 2 fair
**Presentation:** 2 fair
**Contribution:** 2 fair
**Rating:** 5
**Confidence:** 4

**Summary:**

The paper proposes Q-score matching, a new methodology for off-policy reinforcement learning that harnesses the score-based structure of diffusion model policies to align with the action gradients of the Q-function. This approach aims to overcome the limitations of simple behavior cloning in actor-critic settings by integrating the score of the policy with the Q-function's action gradient. Theoretical justification is provided for the proposed method. Effectiveness is demonstrated through comparative experiments in simulated environments.

**Strengths:**

1. The approach of conceptualizing the reinforcement learning problem as a dynamic process is thought-provoking, and employing a diffusion model to ascertain the action progression for each discretization step is an innovative and logical step.

2. A comprehensive theoretical analysis is provided.

**Weaknesses:**

1. **Theory-Practice Disparity** The theoretical framework suggests that the score of optimal policies $\Psi^*$ aligns with the action gradient of the optimal Q-function ($\nabla_a Q^{\Psi^*}$.). While conceptually sound, the challenge arises in practical scenarios where the optimal Q-function is unknown. In your case, you goal is to move $\Psi$ to $\Psi^*$, and you use the $\nabla _a Q$ but not $\nabla_a Q^*$ as the direction to update $\Psi$. The theorem, while insightful, does not appear to significantly contribute to the algorithm's practical efficacy.

2. **Implementation Clarification Needed** There seems to be a discrepancy between the methodology described and its implementation. The paper suggests using a diffusion model to learn the discrete action sequence from $a_{t-1}$ to $a_t$ as indicated in section 2.3. Nonetheless, the algorithm presents the derivation of $a_t$ through iterative denoising from random noise, rather than using $a_{t-1}$ as a starting point. This aspect of the implementation calls for further clarification.

3. **Comparative Methodology Concerns** The implementation closely resembles the one in Diffusion-QL [1], where actions are also sampled by denoising from noise through a trained score function, and this function is refined under the guidance of the Q-value. While Diffusion-QL optimizes the score function by directly maximizing the Q-value, your method minimizes the L2 norm between the score and the action gradient of Q. These methods seem to conceptually converge, raising questions about the distinctiveness of your approach.

4. **Insufficient Experimentation** The range of experiments conducted lacks breadth, especially in the domain of online continuous control tasks where environments like Ant and Humanoid are benchmarks. Incorporating these could enhance the empirical validation of the proposed method.

[1] Wang, Zhendong, Jonathan J. Hunt, and Mingyuan Zhou. "Diffusion policies as an expressive policy class for offline reinforcement learning." arXiv preprint arXiv:2208.06193 (2022).

**Questions:**

1. Confused in the Algorithm box. What is the $\pi_\phi$ in the algorithm? The score network $\Psi$ depends on $(s, a)$. Why the sampling of $a_t$ doesn’t depend on $a_{t-1}$?

---

> ### Author Response · Authors · 2023-11-16
>
> Thank you for your review and for acknowledging the innovation and thought-provoking aspect. Hopefully we can address your concerns, as there seem to be some misconceptions about the paper.
>
> 1. There seems to be a misconception of the main theorem, as we make no dependency on knowing $Q^*$. This is in fact a primary motivation for the theory as described in the 3rd paragraph of the introduction: we want to “score match” to something unknown, so we need to develop new theory that doesn’t rely on knowing anything about $\pi^*$. The main theorem is this, as seen through its contrapositive (and more clearly in its proof): if the score $\Psi$ and action gradient of the Q function with respect to $\Psi$, $\nabla_a Q^\Psi$, are misaligned, then any amount of aligning $\Psi$ to $\nabla_a Q^\psi$ will strictly increase the Q-function towards $Q^*$, and is thus a valid update rule. A large part of this paper's novelty comes from the new math developed to prove this theorem in the appendix, which makes the theoretical role of the QSM algorithm much more clear. We understand the appendix is not required reading, so we have updated the main text to increase clarity.
>
> 2. Note in eq. (6) that the state and action are time-separated systems: for every state step, there are many action steps. For any Langevin dynamics-based method (including diffusion models) to work, there need to be enough steps so that the initialization at each state time step makes little difference. While the state-time initialization $a_{t-1}$ is less important, the internal diffusion steps using $\Psi$ are important, and our theory is applied to develop the QSM algorithm to update $\Psi$ based on these internal steps. *In summary:* initializing between $a_{t-1}$ and Gaussian noise makes little difference, but the latter has some practical benefits, such as the computational graph not  connecting through time, and allowing for bang-bang control, etc.
>
> 3. The main practical difference: all previous works apply diffusion model policies to offline RL problems, while our work focuses on online/off-policy RL. While both try to maximize Q in some way, these are fundamentally different approaches under the hood. For a diffusion model of depth $K$ and denoising model $\phi$, Diffusion-QL needs to backpropagate through $\phi$ composed on itself $K$ times, resembling gradient structures and issues with RNN’s (vanishing/exploding gradients), while QSM only needs to differentiate with respect to $\phi$ itself (for arbitrary depth $K$) – using our theory to utilize the “inner optimization” of diffusion models allows methods like QSM by design to, unlike Diffusion-QL, train well with deeper diffusion models and improve as $K$ is increased, even during training or evaluation. QSM has implicit entropy regularization as described through the pedagogical example in 4.3, while Diffusion-QL needs explicit regularizers. These benefits come from our presented methodology, and the example algorithm QSM, being derived from and respecting the underlying score-based structure of diffusion models, which the direct Q optimization objective does not capture.
>
> 4. The emphasis of this paper is a new theoretical framework to update diffusion model policies, as well as a standard initial implementation in an off-policy  RL setting to demonstrate that the theory works as intended and shows its promise, namely when there are less training samples. Much of this rich theory is developed and presented in the proofs in the appendix; while important for reading, they are quite long and dense, and would greatly hinder readability if crammed in the main body.
> There are plenty of interesting directions to improve on QSM into a state-of-the-art continuous RL algorithm, and many other interesting ways to implement a Q-score matching algorithm (e.g. using $\nabla_a Q$ directly to sample via Langevin dynamics), but we feel the theory is dense and rich enough to deserve its own paper, and position it to lay the foundation for how to use diffusion model structure to design more efficient diffusion policy training algorithms and spark future research. If there are particular insights you hope to gain from Ant and Humanoid, please let us know.
>
> *Questions*: Note $\pi_\phi$ is not modeled explicitly in diffusion models, but we do give how actions are sampled in the first line within the loop. The other questions should be addressed from the above response.
>
> Hopefully this has clarified some points in the paper and addressed your concerns. If so, please consider modifying your scores and rating to reflect this. If not, we are of course happy to continue the discussion.

---

### Author Response · Authors · 2023-11-22
**New experiments on Ant/Quadruped and Humanoid**

As recommended by reviewers, we have included experiments in an appendix section (now Appendix A) including experiments of QSM on both the Ant/Quadruped and Humanoid environments. While the focus of the paper is theoretical, we hope this can better illustrate the practical potential of exploiting the score-based structure of diffusion models for Q-learning.

As a reminder, if our results and our discussion has changed your perspective on the paper, please consider changing your score to reflect this (and please let us know why if not) – there are only 24 hours left to do so. Thank you.

---

### Meta-Review · Area_Chair_n2Fv · 2023-12-21

**Metareview:**

This work proposes a new theoretical framework for utilizing diffusion models for off-policy RL utilizing estimates of the Q function for continuous-time dynamical processes. This new way of performing policy updates has clear theoretical novelty, but the experiments did not meet reviewers' expectations for the field.  I would recommend following some of the reviewers' feedback on potential baselines to consider, when applicable (e.g., standard PGs) and resubmit to an upcoming conference.

**Justification For Why Not Higher Score:**

All reviewers were dissatisfied with experiments

**Justification For Why Not Lower Score:**

N/A

---

### Decision · Program_Chairs · 2024-01-16

Reject